# Large language models are poor clinical administrators: An evaluation of structured queries in real-world electronic health records

Eyal Klang[1]*, Vera Sorin[2], Panagiotis Korfiatis[2], Ashwin S. Sawant[1], Robert Freeman[1], Alexander W. Charney[1], Girish N. Nadkarni[1]*, Benjamin S. Glicksberg[1]*

**1** The Hasso Plattner Institute for Digital Health at Mount Sinai, The Windreich Department of Artificial Intelligence and Human Health, Icahn School of Medicine at Mount Sinai and the Mount Sinai Health System, New York, New York, United States of America, **2** Department of Radiology, Mayo Clinic College of Medicine and Science, Mayo Clinic, Rochester, Minnesota, United States of America

* eyal.klang@mountsinai.org (EK); girish.nadkarni@mountsinai.org (GN); Benjamin.glicksberg@mssm.edu (BG)

## Abstract

Large language models (LLMs) have shown promise in interpreting clinical free-text like provider notes. There is limited evidence on tabular electronic health record (EHR) tasks. Our objective was to evaluate the accuracy of LLMs on structured EHR administrative tasks using direct prompting, chain-of-thought (CoT) reasoning, and tool-enabled code generation. We evaluated nine LLMs randomly sampling from a real-world sampled dataset of 50,000 emergency department (ED) visits. Tasks were tested across 25 combinations of table sizes (5–25 rows and columns). Models were prompted directly or with CoT reasoning to return numerical answers. In the tool setting, models generated Python code, which was executed to retrieve answers. Accuracy was defined as the proportion of model outputs matching validated references. We also assessed JSON format compliance. Across 32,950 model queries, performance varied by model, task type, and prompting strategy. Direct prompting produced uniformly low accuracies. CoT prompting moderately improved performance, particularly for logical filtering, but results degraded significantly as table size increased. The tool-based strategy substantially improved accuracy. Smaller models and distilled reasoning variants had more frequent formatting and execution errors. In conclusion, for structured EHR tabular data extraction, direct and CoT prompting strategies resulted in limited accuracy and poor scalability, particularly as table size increased. Tool-based prompting, where models generated and executed Python code, achieved higher accuracy and valid output formatting. Structured data tasks in clinical workflows may require hybrid approaches that combine LLMs with code execution to ensure accuracy and consistency.

**Data availability statement:** The data cannot be shared publicly due to patient privacy and institutional data protection policies. Access to the data may be granted to qualified researchers upon reasonable request and subject to approval by the Mount Sinai Institutional Review Board, which can be contacted at irb@mssm.edu, for those who meet the criteria for access to confidential information.

**Funding:** The author(s) received no specific funding for this work.

**Competing interests:** The authors have declared that no competing interests exist.

## Author summary

Hospitals depend on electronic health records (EHRs) not only for patient care, but also for everyday administrative tasks such as counting visits, tracking admissions, and identifying groups of patients who meet specific criteria. Large language models (LLMs) are increasingly promoted as tools that could help staff answer these questions using plain language instead of database queries or custom scripts. We tested whether this is currently realistic by evaluating nine LLMs on structured EHR tasks using real emergency department data. The models were asked to count records or filter patients based on one or more conditions, either by answering directly, by using step-by-step reasoning, or by generating Python code to perform the task. Direct answers were often inaccurate, and step-by-step prompting improved performance only modestly. Accuracy also dropped as tables became larger. The most reliable results came from a hybrid approach in which the model generated code and the code was executed to retrieve the answer. These findings suggest that current LLMs are not yet dependable as standalone clinical administrators for structured data tasks, and that safer healthcare use will likely require pairing LLMs with conventional computational tools.

## Introduction

Electronic health records (EHRs) are essential repositories of patient data used for clinical care, hospital operations, and biomedical research [1,2]. Within healthcare settings, administrators routinely perform tasks such as patient census counts, daily patient visits summaries, equipment inventory tracking, and resource utilization monitoring. These tasks are critical for operational planning, quality assurance, billing, and efficient resource allocation [3,4]. However, generating these reports often relies on data analysts. They use structured query language (SQL) or custom scripting to extract relevant data [5]. This reliance introduces operational delays and limits staff's ability to access and interpret information directly from EHR systems when needed [6].

Large language models (LLMs) offer a promising pathway to simplify access to EHR data through their ability to interpret user queries in natural language [7,8]. LLMs have previously demonstrated effectiveness across a variety of clinical applications, including clinical decision support, synthetic patient data generation, clinical question-answering, and empathetic patient communication [9–14]. Additionally, they have been explored for workflow-specific tasks, such as shift change summarization, emergency department documentation, and the creation of administrative dashboards [15,16].

Despite these advances, the use of LLMs for structured clinical administrative queries, such as summarizing patient populations during shift changes, generating daily resource utilization reports, or compiling lists of upcoming patient appointments,

remains underexplored [17]. These structured tasks require precise logical reasoning, consistent numerical computation, and accurate data filtering, capabilities that many LLMs have yet to reliably demonstrate [18].

Prior research mainly focused on using LLMs to extract structured information from clinical free text, such as pathology or radiology reports [19–21]. Few studies have directly compared multiple LLMs on standardized administrative tasks [22]. To address this gap, we conducted a head-to-head evaluation of nine LLMs, utilizing direct prompting, chain-of-thought (CoT), and a tool-enabled approach (i.e., LLMs generating executable code for data queries) to better understand their out-of-the-box performance on structured EHR administrative tasks. Our goal was to systematically quantify model accuracy, identify critical areas of failure, and provide clear guidance for leveraging LLMs effectively and safely in healthcare administrative workflows.

## Methods

### Study design and setting

We evaluated nine LLMs on two tabular tasks, counting and logical filtering, in a real-world EHR environment. We also tested a tool-based approach in which LLMs generated Python code to perform the same filtering tasks, allowing us to run the code against a known reference and compare outputs. Further study details can be found in the TRIPOD-LLM checklist [23] (S1 Checklist).

### Data source and IRB statement

The Mount Sinai Hospital Institutional Review Board approved the study. We extracted 407,080 emergency department (ED) visits from 2024 via the Mount Sinai Health System's data warehouse, which encompasses seven hospitals in New York City. From these notes we randomly selected 50,000. Each note included 93 complete entries (See S1 List). Records with missing data were excluded.

### LLM models and prompting

We evaluated each model in two ways: an "out-of-the-box" approach and a tool approach (Table 1). The out-of-the-box approach included direct and CoT prompting. We also tested distilled models that emulate CoT internally. In the out-of-the-box approach, we directly provided tables in CSV format and prompted the models to return counts or apply logical filters. In the tool approach, the models were instructed to generate executable Python code. We then executed the code and compared the outputs against the same tables to test accuracy.

**"Out-of-the-box" counting and logical filtering.** In the counting task, models were instructed to return the number of rows that met a single condition (S1–S2 Prompts for direct and CoT versions). In the logical filtering task, models were instructed to return the number of rows that met multiple conditions, including one ro more exclusion criteria (S3–S4 Prompts).

Five models were tested with both direct ("simple") prompts and CoT prompts. The remaining four models, namely DeepSeek reasoning versions, incorporate reasoning internally and were tested using direct prompts only. For both tasks, the input table was included in the prompt in CSV format. The output from each was compared to a ground-truth reference computed via Python.

**Tool-enabled approach.** For the tool approach, each model was prompted to produce executable Python code to perform the logical filtering task (S5 Prompt). We used a Python function to parse and execute each LLM-generated code snippet in an isolated environment, compiling the code and running it under controlled globals and locals. We compared the output against the same reference DataFrame to verify the results. The complete list of all experiments and the number of API calls per model is detailed in Table 2.

### Dataset construction

We created tables by sampling from the 50,000-row dataset, creating 25 combinations (5, 10, 15, 20, or 25 rows × 5, 10, 15, 20, or 25 columns). For each combination we created 50 unique tables through random sampling. This process was

PLOS Digital Health

**Table 1. Large Language Models Evaluated in the Study.**

| Company | Access Type | Model | Version/ HuggingFace | Parameters | Context Window | Access |
|---|---|---|---|---|---|---|
| openAI | Closed | GPT-4o | gpt-4o-2024-08-06 | NA | 128K | Azure Tenant |
| Meta | Open | Llama-3.1-8B | meta-llama/ Llama-3.1-8B-Instruct | 8B | 128K | On-Prem |
| Meta | Open | Llama-3.3-70B | meta-llama/ Llama-3.3-70B-Instruct | 70B | 128K | On-Prem |
| Meta/ DeepSeek (finetune) | Open | DeepSeek-R1-Distill-Llama-8B | deepseek-ai/ DeepSeek-R1-Distill-Llama-8B | 8B | 128K | On-Prem |
| Meta/ DeepSeek (finetune) | Open | DeepSeek-R1-Distill-Llama-70B | deepseek-ai/ DeepSeek-R1-Distill-Llama-70B | 70B | 128K | On-Prem |
| Alibaba | Open | Qwen-2.5-7B | Qwen/Qwen2.5-7B-Instruct | 7B | 128K | On-Prem |
| Alibaba | Open | Qwen-2.5-72B | Qwen/Qwen2.5-7B-Instruct | 72B | 128K | On-Prem |
| Alibaba/ DeepSeek (finetune) | Open | DeepSeek-R1-Distill-Qwen-7B | deepseek-ai/ DeepSeek-R1-Distill-Qwen-7B | 7B | 128K | On-Prem |
| Alibaba/ DeepSeek (finetune) | Open | DeepSeek-R1-Distill-Qwen-72B | deepseek-ai/ DeepSeek-R1-Distill-Qwen-72B | 72B | 128K | On-Prem |

Details of the nine large language models (LLMs) evaluated, including model versions, parameter counts, context window size, and access method.

**Table 2. Experimental Tasks, Prompting Strategies, and API Calls.**

| Task | Prompt Type | Experiments | API calls |
|---|---|---|---|
| Count | Direct Prompt | 5..25 rows, 5..25 cols, 50 experiments per combination | 1250 * 9 models = 11,250 |
| Count | CoT Prompt | 5..25 rows, 5..25 cols, 50 experiments per combination | 1250 * 4 models = 5,000 |
| Logical Filter | Direct Prompt | 5..25 rows, 5..25 cols, 50 experiments per combination | 1250 * 9 models = 11,250 |
| Logical Filter | CoT Prompt | 5..25 rows, 5..25 cols, 50 experiments per combination | 1250 * 4 models = 5,000 |
| Code Logical Filter | Tool | 50 experiments | 50 * 9 models = 450 |
| | | | Total 32,950 |

Summary of experimental tasks (counting, logical filtering, and tool-based logical filtering), prompting methods, namely direct, Chain-of-Thought (CoT), and tool approach, and total API calls performed across models.

repeated for all nine models, resulting in 25 sets of 50 samples per model. This process resulted in overall 32,950 discrete model queries across various table dimensions, tasks, and prompt combinations. An overview of the study workflow is illustrated in Fig 1.

## Sample size and power

A one-way ANOVA power analysis with five groups ($\alpha = 0.05$, power $= 0.80$) and a hypothesized 10% difference as a large effect size (Cohen's $f = 0.5$) indicated the need for 50 replicates per group (via the FTestAnovaPower function in *statsmodels*). We therefore repeated each row-column configuration 50 times for each model.

## Statistical analysis

The main outcome was accuracy, defined as the proportion of counts or filters matching a validated reference. When Python code was generated, its output was also checked against the same reference. Secondary outcomes included JSON formatting validity and code syntax correctness. We used descriptive statistics to compare performance across

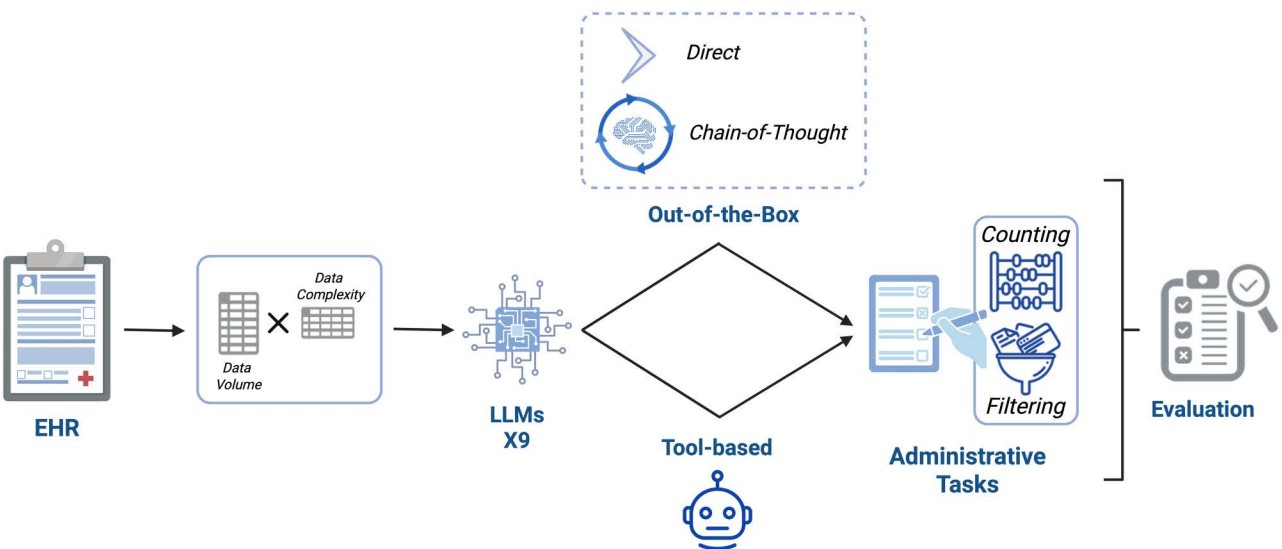

**Fig 1. Overview of Study Workflow.** Schematic depicting the evaluation pipeline for large language models (LLMs) on structured administrative tasks using electronic health record (EHR) data. Input tables varied in volume (row count) and complexity (column count). Nine LLMs were tested using either direct or chain-of-thought (CoT) prompting ("out-of-the-box") or a tool-enabled strategy that generated executable code. Tasks included counting and logical filtering. Model outputs were evaluated against reference answers for accuracy and formatting compliance.

row-column dimensions, prompt types, and model classes. Confidence intervals were derived using the Wilson score interval for binomial proportions. Between-group differences in accuracy (proportion correct) were tested using two-sided two-proportion z-tests with pooled standard errors ($\alpha = 0.05$).

## Computational infrastructure

All GPT-based queries were run through Mount Sinai Hospital's Azure tenant, a HIPAA-compliant environment. Experiments with open-source models were conducted on a local cluster with four H100 GPUs. Default hyperparameters were used for each model. Data processing and analyses were done with Python 3.9.18, using *torch*, *transformers*, *NumPy*, and *pandas*.

## Results

### Cohort characteristics

Of the ED visits extracted, most patients were triaged as urgent (emergency severity index [ESI] 3, 58.8%), followed by emergent (ESI 2, 19.7%), and less urgent (ESI 4, 17.5%) (S1 Table). Most patients arrived by personal means (66.2%), or by emergency transport (30.3%) (S2 Table). The majority were discharged following their visit (75.2%), and 16.1% were admitted (S3 Table).

### JSON compliance and output quality

Models exhibited substantial variability in their ability to produce valid JSON output across tasks and prompt strategies (Table 3). Llama-3.1-8B demonstrated the highest failure rates, with JSON formatting errors in 96.0% of direct count tasks, 99.6% of direct filter tasks, and 84.0% of tool-based filtering tasks. Due to these consistently high failure rates, the model was excluded from further quantitative analysis. Llama-3.3-70B performed well in both count and filter tasks using direct and CoT prompting (0.0–0.08% failure), but surprisingly failed to return structured JSON in 98.0% of tool-style filtering tasks,

**Table 3. JSON Output Failure Rates by Model, Task, and Prompt Type.**

| Model | Prompt Strategy | Task | Failure Rate |
|---|---|---|---|
| DeepSeek-R1-Distill-Llama-70B | Direct | Count | 0.32% |
| DeepSeek-R1-Distill-Llama-70B | Direct | Filter | 0.32% |
| DeepSeek-R1-Distill-Llama-70B | Tool | Filter | 34.0% |
| DeepSeek-R1-Distill-Llama-8B | Direct | Count | 0.32% |
| DeepSeek-R1-Distill-Llama-8B | Direct | Filter | 0.48% |
| DeepSeek-R1-Distill-Llama-8B | Tool | Filter | 48.0% |
| DeepSeek-R1-Distill-Qwen-7B | Direct | Count | 0.08% |
| DeepSeek-R1-Distill-Qwen-7B | Direct | Filter | 0.08% |
| DeepSeek-R1-Distill-Qwen-7B | Tool | Filter | 50.0% |
| GPT4o | CoT | Count | 0.0% |
| GPT4o | CoT | Filter | 0.0% |
| GPT4o | Direct | Count | 0.0% |
| GPT4o | Direct | Filter | 0.0% |
| GPT4o | Tool | Filter | 2.0% |
| Llama-3.1-8B | CoT | Count | 32.32% |
| Llama-3.1-8B | CoT | Filter | 47.36% |
| Llama-3.1-8B | Direct | Count | 96.0% |
| Llama-3.1-8B | Direct | Filter | 99.6% |
| Llama-3.1-8B | Tool | Filter | 84.0% |
| Llama-3.3-70B | CoT | Count | 0.08% |
| Llama-3.3-70B | CoT | Filter | 0.08% |
| Llama-3.3-70B | Direct | Count | 0.0% |
| Llama-3.3-70B | Direct | Filter | 0.0% |
| Llama-3.3-70B | Tool | Filter | 98.0% |

Rates of JSON formatting failures across evaluated LLMs, task types, and prompting strategies, highlighting the variability and specific prompt-related issues among different models.

indicating difficulty with code-wrapped outputs despite strong overall reasoning capabilities. In contrast, GPT-4o and the Qwen-2.5 models (7B and 72B) exhibited near-perfect JSON compliance across all tasks and prompt strategies, with failure rates of 0.0–0.4%, including under tool-enabled prompting. These models consistently returned well-structured outputs, regardless of the complexity of the task. The DeepSeek-distilled models showed moderate formatting reliability in direct prompting (≤0.5% failure) but consistently struggled in tool settings, where failure rates ranged from 34.0% to 50.0%. These issues stemmed largely from malformed JSON structures or failure to wrap generated code in the correct output schema.

### Effectiveness and limitations of prompting strategies

Model performance was positively correlated with size (parameter count), with larger models generally outperforming smaller and distilled variants (Table 4). However, performance varied substantially based on prompting strategies. While CoT generally improved accuracy compared to direct prompting, its effectiveness was notably task-dependent and model-specific.

GPT-4o accuracy improved from 68.6% with direct prompting to 77.6% ($p < 0.001$) using CoT in counting tasks, and from 49.4% to 77.4% ($p < 0.001$) in logical filtering tasks. Qwen-2.5-72B similarly benefitted from CoT prompting, showing significant gains from 47.0% to 72.3% ($p < 0.001$) in counting and 35.2% to 73.1% ($p < 0.001$) in filtering tasks. However, despite these gains, both direct and CoT prompting still left substantial room for error, especially as dataset

**Table 4. Accuracy of LLMs by Task and Prompting Strategy.**

| Model | Task | Prompt Type | Accuracy (95% CI) |
|---|---|---|---|
| DeepSeek-R1-Distill-Llama-70B | count | Direct | 63.3% (95% CI 60.6% - 66.0%) |
| DeepSeek-R1-Distill-Llama-70B | filter | Direct | 60.4% (95% CI 57.6% - 63.0%) |
| DeepSeek-R1-Distill-Llama-8B | count | Direct | 37.0% (95% CI 34.4% - 39.7%) |
| DeepSeek-R1-Distill-Llama-8B | filter | Direct | 28.9% (95% CI 26.4% - 31.4%) |
| DeepSeek-R1-Distill-Qwen-7B | count | Direct | 34.4% (95% CI 31.8% - 37.1%) |
| DeepSeek-R1-Distill-Qwen-7B | filter | Direct | 20.3% (95% CI 18.1% - 22.6%) |
| GPT4o | count | CoT | 77.6% (95% CI 75.2% - 79.8%) |
| GPT4o | count | Direct | 68.6% (95% CI 66.0% - 71.2%) |
| GPT4o | filter | CoT | 77.4% (95% CI 75.0% - 79.7%) |
| GPT4o | filter | Direct | 49.4% (95% CI 46.7% - 52.2%) |
| Qwen-2.5-72B | count | CoT | 72.3% (95% CI 69.8% - 74.7%) |
| Qwen-2.5-72B | count | Direct | 47.0% (95% CI 44.2% - 49.7%) |
| Qwen-2.5-72B | filter | CoT | 73.1% (95% CI 70.6% - 75.5%) |
| Qwen-2.5-72B | filter | Direct | 35.2% (95% CI 32.6% - 37.9%) |
| Qwen-2.5-7B | count | CoT | 42.4% (95% CI 39.7% - 45.2%) |
| Qwen-2.5-7B | count | Direct | 23.1% (95% CI 20.9% - 25.5%) |
| Qwen-2.5-7B | filter | CoT | 35.8% (95% CI 33.2% - 38.5%) |
| Qwen-2.5-7B | filter | Direct | 21.9% (95% CI 19.7% - 24.3%) |
| Llama-3.3-70B | count | CoT | 70.4% (95% CI 67.8% - 72.8%) |
| Llama-3.3-70B | count | Direct | 51.4% (95% CI 48.6% - 54.1%) |
| Llama-3.3-70B | filter | CoT | 85.8% (95% CI 83.8% - 87.7%) |
| Llama-3.3-70B | filter | Direct | 46.1% (95% CI 43.3% - 48.9%) |

Overall accuracy rates (with 95% confidence intervals) for each model across counting and logical filtering tasks, differentiated by prompting strategy (direct vs. Chain-of-Thought [CoT]).

size increased, indicating limitations in scalability and computational reasoning capability even among larger models (**Figs 2-4**).

## Failures of chain-of-thought and direct prompting

Analysis of failures revealed distinct patterns associated with direct and CoT prompting strategies. Direct prompting frequently resulted in lower baseline performance across all models, particularly in complex filtering tasks (Fig 3, panels A-B). Models often failed to capture multi-step logic or consistently apply exclusion criteria correctly. CoT prompting was designed to mitigate such issues by providing explicit intermediate reasoning steps. Nonetheless, CoT strategies were only partially successful, especially when the number of records (rows) increased significantly. As shown in Fig 3 (panels C-D), model accuracy notably degraded when moving from smaller (5–10 rows) to larger datasets (20–25 rows). Even GPT-4o, the highest performing model, exhibited accuracy decreases from ~95% at minimal dataset sizes to below 60% at larger dataset sizes under CoT conditions.

Additionally, CoT prompting was less effective in scenarios requiring significant numerical aggregation or sustained memory of intermediate results across multiple records. This performance degradation was particularly acute for smaller models, such as Qwen-2.5-7B, where accuracy rapidly declined from approximately 80% to below 20% with increasing dataset size (Fig 2).

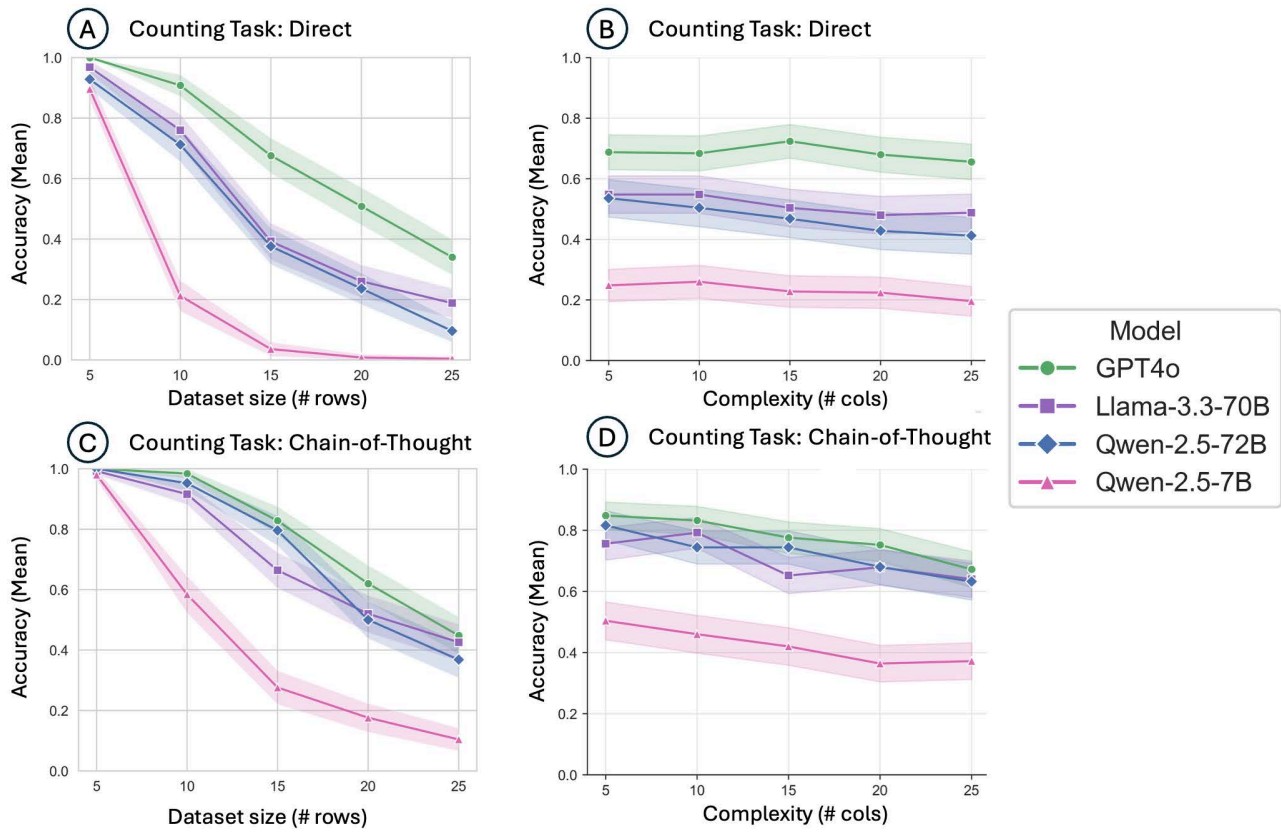

**Fig 2. Model performance on the Counting task across dataset size and complexity, comparing direct and chain-of-thought prompting.** Mean accuracy of GPT4o, Llama-3.3-70B, Qwen-2.5-72B, and Qwen-2.5-7B models using direct prompting. (A) Mean accuracy across dataset sizes (number of rows); (B) Mean accuracy across complexity levels (number of columns); (C) Mean accuracy across dataset sizes (number of rows) using chain-of-thought (CoT) prompting; (D) Mean accuracy across complexity levels (number of columns) using CoT prompting.

## Impact of dataset size vs. complexity

Distinct effects of dataset size (number of records) versus complexity (number of data fields) were observed (Figs 2-4). Increasing dataset size had a markedly negative impact across all models and task types, reflecting limitations in maintaining accurate numeric aggregation and logical consistency across larger volumes of data. In contrast, increasing data complexity by adding more fields (columns) had a subtler effect, slightly decreasing performance without causing the same magnitude of decline observed with increased dataset size. This suggests that numeric reasoning and aggregation present greater computational challenges than multi-condition logic for current LLMs.

## Performance of tool-enabled strategy for filtering task

The tool strategy, in which models were prompted to generate executable Python code to perform the filtering task, demonstrated substantial variation in performance across models. GPT-4o and Qwen-2.5 models exhibited excellent accuracy, achieving nearly perfect performance with minimal JSON compliance errors (2% and 0%, respectively; Table 3, Fig 5). In contrast, distilled DeepSeek variants struggled significantly under this approach, demonstrating markedly lower accuracy accompanied by high JSON compliance failure rates ranging from 34% to 50%. The primary source of these errors involved syntactic mistakes or incorrect JSON formatting that rendered code non-executable, underscoring a critical limitation in their capacity to reliably operate as independent agents for structured data extraction tasks.

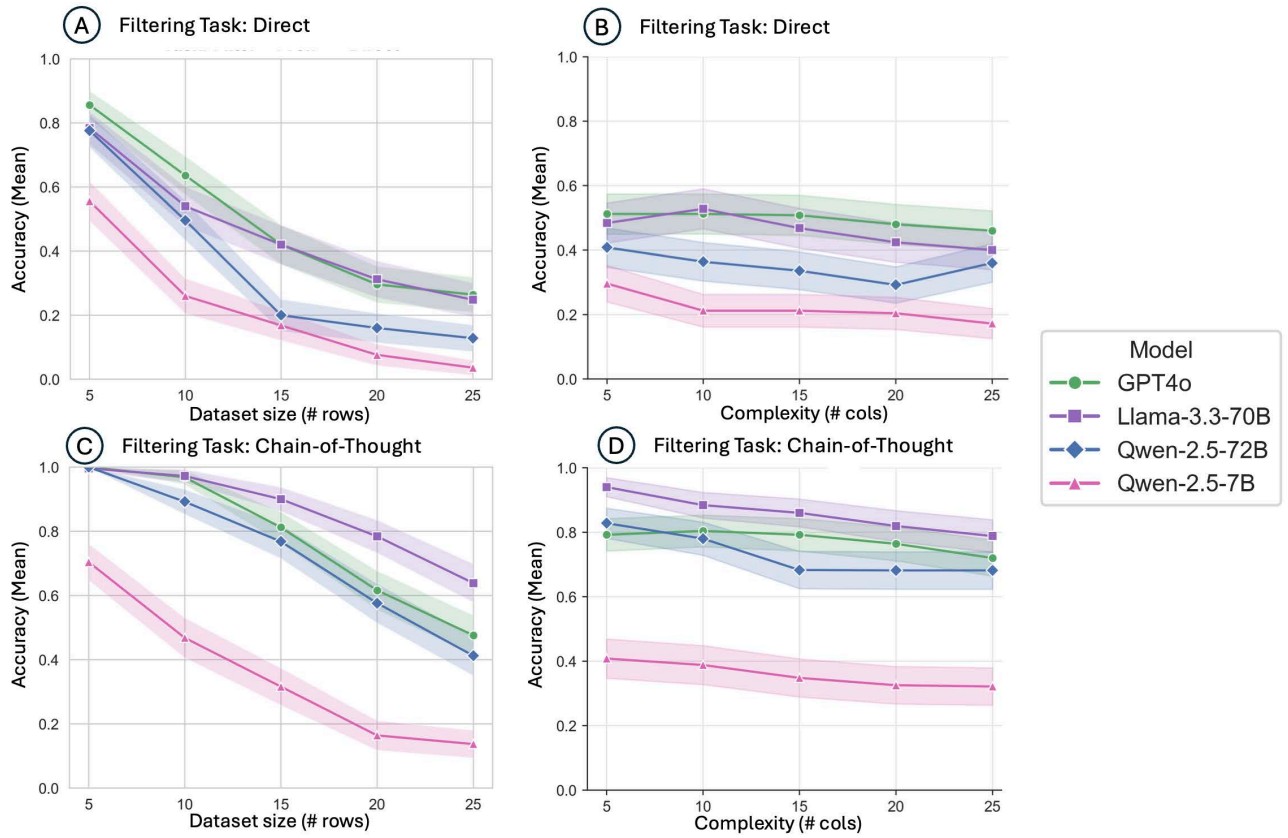

**Fig 3. Model performance on the Filtering task across dataset size and complexity, comparing direct and chain-of-thought prompting.** Mean accuracy of GPT4o, Llama-3.3-70B, Qwen-2.5-72B, and Qwen-2.5-7B models using direct prompting. (A) Mean accuracy across dataset sizes (number of rows); (B) Mean accuracy across complexity levels (number of columns); (C) Mean accuracy across dataset sizes (number of rows) using chain-of-thought (CoT) prompting; (D) Mean accuracy across complexity levels (number of columns) using CoT prompting.

## Comparative evaluation of direct, chain-of-thought, and tool strategies

When comparing across the three strategies (direct, CoT, and tool-based) for the filtering task, the tool-based approach generally showed superior performance, particularly among the largest and most advanced models (GPT-4o, Qwen-2.5-72B). Direct prompting consistently produced the lowest accuracies, especially with increasing dataset complexity and size (Fig 3), while the CoT strategy provided modest but significant improvements for tasks requiring complex logical reasoning or multi-step data filtering (Fig 3C and 3D). Reasoning models, designed to internally emulate chain-of-thought without explicit prompting, failed to match the performance of models explicitly guided through CoT or tool-enabled strategies.

## Discussion

This study demonstrates that current LLMs, when used directly for structured administrative queries in EHR, consistently exhibit substantial limitations, even with small datasets (e.g., ≤ 25 rows). Accuracy and format compliance varied significantly across models, tasks, and prompting strategies. Notably, even the highest-performing models such as GPT-4o struggled to achieve acceptable accuracy levels when relying on direct or CoT prompts alone, particularly as query complexity or dataset size increased.

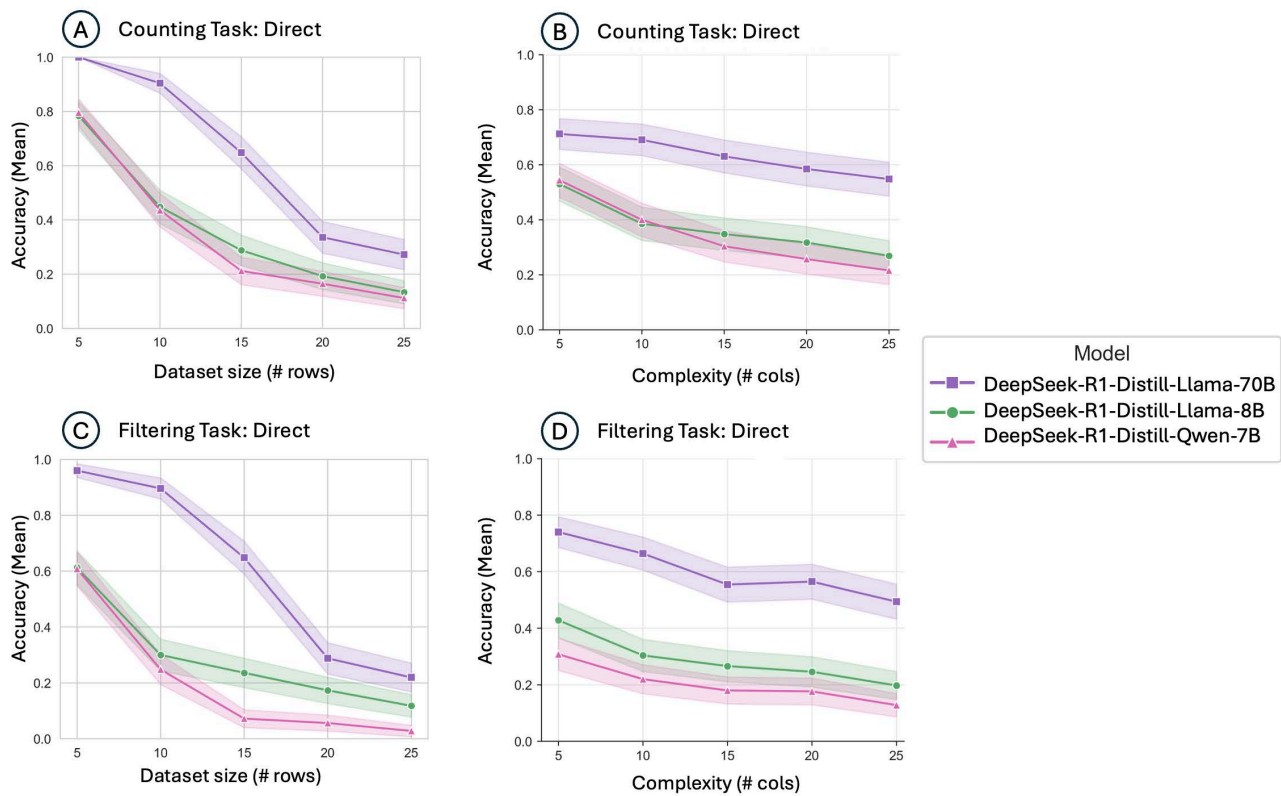

**Fig 4. Performance of distilled models on Counting and Filtering Tasks using direct prompting.** Mean accuracy DeepSeek-R1-Distill-Llama-70B, DeepSeek-R1-Distill-Llama-8B, and DeepSeek-R1-Distill-Qwen-7B models using direct prompting. (A) Mean accuracy across dataset sizes (number of rows) for the Counting Task among distilled models; (B) Mean accuracy across complexity levels (number of columns) for the Counting Task; (C) Mean accuracy across dataset sizes (number of rows) for the Filtering Task; (D) Mean accuracy across complexity levels (number of columns) for the Filtering Task.

LLMs have been previously reported to struggle with tabular data. Previous efforts for LLMs' tabular data analysis in healthcare mainly focused on data extraction, including benchmarks such as EHRSQL [24,25] and MIMICSQL [26,27]. Our study systematically analyzes how nine different models analyze tabular data under stress conditions with increasing data complexity, and under various prompting condition, a gap that was yet to be addressed in the literature. Some commercial tools, such as Google's Healthcare Data Search, now incorporate LLMs for structured data retrieval [28]. Yet, there has been little evidence on how LLMs perform when applied directly to structured healthcare data, particularly with larger datasets, complex logic, or strict formatting requirements.

In our study, direct prompting strategies resulted in uniformly low accuracies across all models for both counting and filtering tasks, highlighting the insufficiency of straightforward natural language instructions for structured data extraction tasks. CoT prompting yielded moderate improvements, particularly in scenarios involving logical filtering tasks; however, these gains were inconsistent and insufficient for practical, reliable clinical use. Distilled reasoning models, intended to internally emulate CoT processes, similarly underperformed, indicating that implicit reasoning alone does not guarantee performance gains.

The tool-enabled approach, where LLMs generated executable Python code to perform logical filtering, emerged as the only viable strategy, markedly improving accuracy and reliability in advanced models (GPT-4o and Qwen-2.5-72B). This finding aligns with results from Shi et al. publication on EHRAgent [29], which showed that code-generating agents

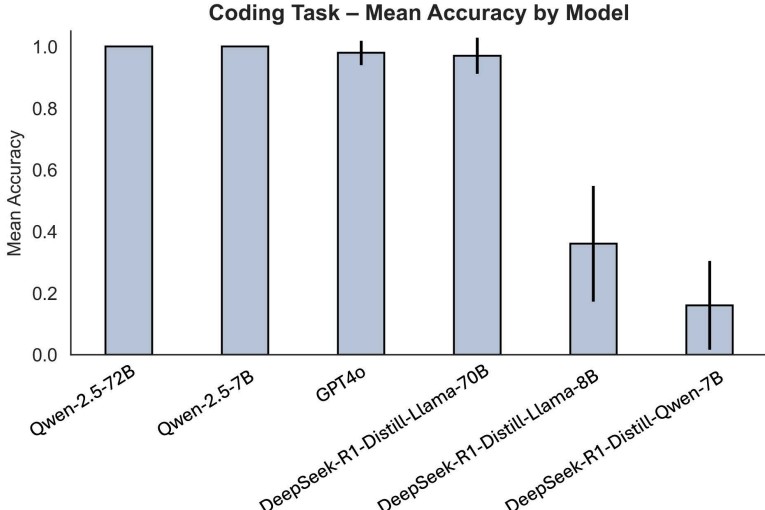

**Fig 5. Coding Task mean accuracy by model.** Mean accuracy (+/- standard error) for models Qwen-2.5-72B, Qwen-2.5-7B, GPT4o, DeepSeek-R1-Distill-Llama-70B, DeepSeek-R1-Distill-Qwen-7B, and DeepSeek-R1-Distill-Llama-8B.

outperform both direct and CoT-based approaches. This highlights that current LLMs can effectively interpret the intent behind structured queries, but still struggle with reliable, basic data pulls without help in this form. Across all tasks, format compliance (valid JSON output) was problematic, especially among smaller or distilled models, severely limiting their practical integration into automated clinical workflows. The widespread inability of models to consistently provide structured and syntactically correct outputs, even with explicit formatting instructions, underscores a major deployment barrier.

Our findings indicate that without explicit computational delegation (i.e., a tool-based strategy), current LLMs are unsuitable for standalone use even on minimally complex administrative tasks in clinical settings. Their frequent errors, inability to express uncertainty, and formatting inconsistencies introduce substantial risks for clinical and administrative operations, particularly in high-stakes scenarios involving billing, compliance, or resource allocation [30,31].

This study has its limitations. Models did not directly access databases or receive iterative feedback, potentially restricting generalizability and requiring further operational efforts. Additionally, we did not assess runtime performance or cost-efficiency, factors crucial for practical deployment. Future studies could potentially address these gaps through fine-tuning models specifically for structured EHR tasks, implementing confidence or verification systems, and evaluating hybrid workflows in real-world clinical contexts.

In conclusion, despite their potential in natural language interpretation, current LLMs require integration with explicit computational tools (such as code-generation approaches) to reliably execute even basic structured queries against healthcare data. Hybrid frameworks combining LLMs with traditional database or coding strategies, like the Model Context Protocol (MCP) [32], are currently required to achieve practical utility, accuracy, and safety in clinical administration tasks.

## Supporting information

**S1 Checklist. TRIPOD-LLM checklist.**
(PDF)

**S1 List. EdEncounterFact Table Columns Names.**
(DOCX)

**S1 Table. Emergency Severity Index (ESI) Acuity Level Distribution.**
(DOCX)

**S2 Table. Patient Arrival Method Distribution.**
(DOCX)

**S3 Table. Emergency Department Disposition Distribution.**
(DOCX)

**S1 Prompt. Counting Task (Direct Prompt).**
(DOCX)

**S2 Prompt. Counting Task (Chain-of-Thought Prompt).**
(DOCX)

**S3 Prompt. Logical Filtering Task (Direct Prompt).**
(DOCX)

**S4 Prompt. Logical Filtering Task (Chain-of-Thought Prompt).**
(DOCX)

**S5 Prompt. Logical Filter Task (Agentic Strategy).**
(DOCX)

## Author contributions

**Conceptualization:** Eyal Klang, Alexander W Charney, Girish N Nadkarni, Benjamin Glicksberg.

**Data curation:** Benjamin Glicksberg.

**Formal analysis:** Eyal Klang, Benjamin Glicksberg.

**Investigation:** Eyal Klang, Vera Sorin, Panagiotis Korfiatis, Ashwin S Sawant, Alexander W Charney, Girish N Nadkarni, Benjamin Glicksberg.

**Methodology:** Eyal Klang, Vera Sorin, Panagiotis Korfiatis, Ashwin S Sawant, Alexander W Charney, Girish N Nadkarni, Benjamin Glicksberg.

**Resources:** Girish N Nadkarni, Benjamin Glicksberg.

**Software:** Eyal Klang, Benjamin Glicksberg.

**Supervision:** Girish N Nadkarni.

**Validation:** Eyal Klang, Ashwin S Sawant, Benjamin Glicksberg.

**Visualization:** Vera Sorin.

**Writing – original draft:** Benjamin Glicksberg.

**Writing – review & editing:** Eyal Klang, Vera Sorin, Panagiotis Korfiatis, Ashwin S Sawant, Robert Freeman, Alexander W Charney, Girish N Nadkarni.

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
